# Post-Fire Seismic Property of Reinforced Concrete Frame Joints with Carbon Fiber-Reinforced Polymer Using Numerical Analysis

**Jinyan Wang, Xingchao Yue, Fuli Li** **, Yuzhou Sun \* and Ziqi Li**

School of Civil Engineering and Architecture, Zhongyuan University of Technology, Zhengzhou 450007, China; 5440@zut.edu.cn (J.W.); 2016109135@zut.edu.cn (X.Y.); lifuli@zut.edu.cn (F.L.); 2021109355@zut.edu.cn (Z.L.)
\* Correspondence: sunyz@zut.edu.cn; Tel.: +86-135-2356-1916

**Abstract:** This study investigated the post-fire seismic characteristics of reinforced concrete frame joints with carbon fiber-reinforced polymer (CFRP) under low-cycle reciprocating loads through numerical analysis. Finite element simulations were conducted to examine the hysteretic curve, skeleton curve, energy dissipation, and stress distribution of the reinforced joints. The findings revealed that, relative to unreinforced joints post-fire, the bearing capacity of the reinforced joints remained essentially unaltered during the elastic phase. However, their ultimate bearing capacity, energy dissipation capacity, and ductility exhibited varying degrees of enhancement. Interestingly, this augmentation did not persist as the number of reinforcement layers increased. The optimal reinforcing effect was observed with the application of two reinforcement layers, resulting in a 30.3% increase in ultimate bearing capacity and a 26.5% improvement in energy dissipation capacity. Moreover, as the axial compression ratio increased, the high-stress zones within the joint expanded, and the failure mode transitioned from plastic damage at the beam end of the joint under low axial compression ratios to column crushing failure under high axial compression ratios.

**Keywords:** reinforcement; frame joint; after fire; seismic performance; numerical analysis

## 1. Introduction

As economic and technological development progresses, the propensity for fire-causing factors escalates, thereby increasing the frequency of building fires and complicating fire prevention and control measures. Recent studies [1–8] revealed substantial degradation in the bearing properties, stiffness, and strength of joints following fire exposure. Experimental research by Li et al. [7,8] demonstrated a marked decrease in the bearing capacity of joints post-fire. After 75 min of fire exposure, joint bearing capacity can decrease by 13.2–15.8%, and after 120 min, it can decrease by 33.1–34.9%.

Despite these effects, most reinforced concrete structures maintain a certain level of bearing capacity and can continue to be used after appropriate reinforcement. Therefore, reinforcing and repairing fire-damaged buildings not only ensures their usability, but also aligns with China's sustainable development strategy.

Various repair methods are commonly used in construction engineering [9–12] to restore damaged structures, enhance seismic resistance, and improve durability. Methods such as enlarging the cross-section, cladding steel, bonding steel, bonding fiber, base-isolation, and anti-buckling bracing are typically used to strengthen reinforced concrete frame structures. Each of these methods possesses unique advantages and disadvantages, and they were applied and studied extensively in architectural practice. The joint area in frame structures plays a critical role in transmitting and distributing internal forces, ensuring the structure's integrity. It was found that the joint area is the most vulnerable region of a frame structure to damage. Despite numerous studies on the post-fire residual bearing properties and seismic performance of structures such as reinforced concrete

beams, columns, and shear walls [13–15], and significant progress in strengthening room-temperature reinforced concrete structures [16–20], limitations in testing facilities and methods impeded comprehensive performance studies.

Carbon fiber, a novel fiber composite material with over 95% carbon content, exhibits high strength, high elastic modulus, and an impressive strength-to-weight ratio [21–25], making it ideal for joint reinforcement. Therefore, it is crucial from a theoretical standpoint to evaluate the post-fire mechanical performance of reinforced concrete beam-column joints and propose appropriate repair and reinforcement solutions.

Based on the mechanics model theory for reinforced concrete frames, this study employs the finite element method to investigate the post-fire seismic properties of carbon fiber-reinforced joints using numerical simulation. The seismic performance of the joints is assessed using various metrics, including hysteretic curves, skeleton curves, bearing capacity, energy dissipation, ductility, and stress distribution. Our goal is to evaluate the seismic resistance of joints under different reinforcement conditions, providing a comprehensive analysis of the results.

## 2. Specimen Design

The dimensions of the joint model for this study were based on the seismic experiments conducted on Xue's reinforced concrete frame joints [13], which served to verify the accuracy of the numerical simulation results presented in this paper. A detailed depiction of the reinforced concrete frame joint model is presented in Figure 1. The design reinforcement and sectional sizes of the beam-column joint are summarized in Table 1, while the mechanical properties of the reinforcement are detailed in Table 2. The compressive strength of the concrete cube was measured at 31.9 MPa, with an elastic modulus of 3300 MPa.

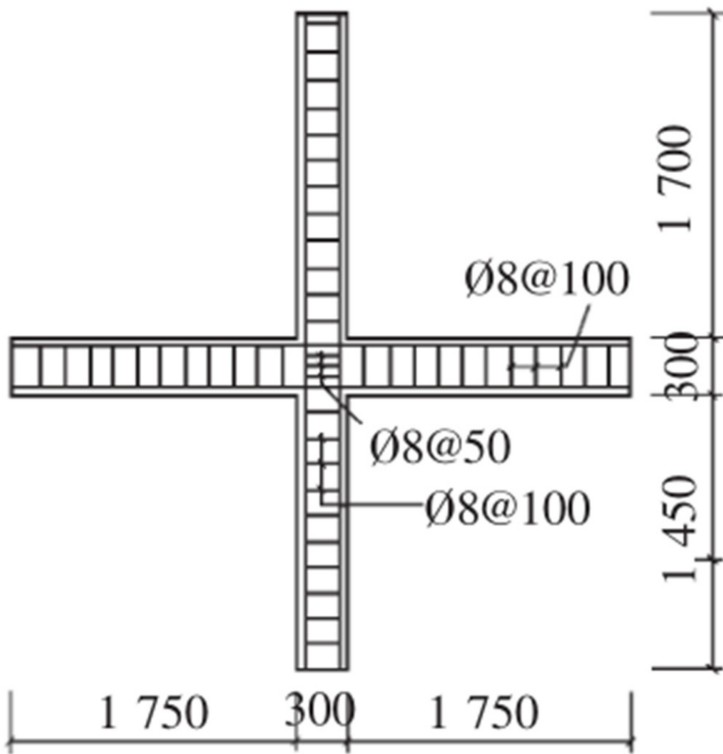

**Figure 1.** Details of the joint (unit: mm).

**Table 1.** Joint reinforcement.

|  | Joint |
|---|---|
| Concrete strength grade | C30 |
| Longitudinal reinforcement of the beam | 6φ16 |
| Longitudinal reinforcement of the column | 6φ20 |
| Stirrups of the beam | φ8@100 |
| Stirrups of the column | φ8@100 |
| Stirrups of the core area | φ8@50 |
| Beam sectional size (mm·mm) | 200 × 300 |
| Column sectional size (mm·mm) | 300 × 300 |
| Protective layer thickness (mm) | 30 |
| Axial compression ratio | 0.3 |

**Table 2.** Mechanical nature of the reinforcement.

| Reinforcement Diameter/mm | Elastic Modulus/MPa | Yield Strength/MPa |
|---|---|---|
| 20 | $1.99 \times 10^5$ | 484 |
| 16 | $2.05 \times 10^5$ | 452.5 |

## 3. Simulation of Temperature Field

The fire exposure duration in the experiment was set at 60 min, with fire applied to all four sides of the specimen. The specific temperature rise and fall curve can be seen in Figure 2.

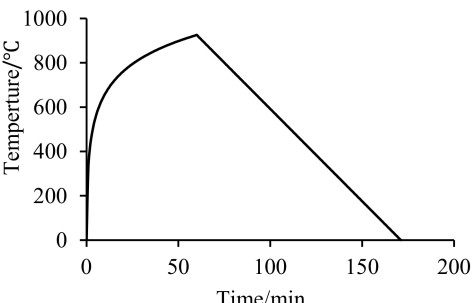

**Figure 2.** Temperature rise and fall curve.

In accordance with the fundamental principles of heat transfer, this study utilized nonlinear finite element analysis to investigate the beam's transient temperature field [26–28] and constructed a three-dimensional model of the reinforced concrete frame joint. It was determined that the reinforcement had negligible effects on the temperature field and, as such, was not considered in this study. The temperature of the concrete where the reinforcement was located was approximated as the reinforcement temperature.

The thermal parameters of the materials, such as the density, heat transfer coefficient, specific heat capacity, and expansion coefficient of the reinforcement and concrete, were calculated using the corresponding specifications and formulas from Eurocode 3 for simulating the temperature field [29].

Concerning the interaction, the beams and columns were subjected to fire on all sides, with the ISO-834 standard fire curve, proposed by the International Organization for Standardization, used as the temperature amplitude curve [30,31]. The initial temperature of the predefined field was 20 °C, the convective heat transfer coefficient was 25 W/(m²·°C), the integrated radiation coefficient was 0.7, the absolute zero was −273 °C, and the Boltzmann constant was $5.67 \times 10^{-8}$ W/(m²·k⁴).

The division of finite elements impacts the convergence and accuracy of the calculation results. Generally, the finer the mesh division, the more accurate the calculation results will be, and the easier it will be to converge. However, this places high demands on equipment performance and significantly increases calculation time. Bearing these factors in mind, this paper adopted a seed density of 50 mm, a DC3D8 element for concrete, and a DC1D2 element for reinforcement, determined through trial calculations.

Figure 3 presents the maximum temperature at different locations of the post-fire (*t* = 60 min) joint. The temperature fields of the beam and column sections exhibit symmetry with respect to the longitudinal and transverse axes of the corresponding cross-section. Heat was transferred layer by layer from the outermost layer of the reinforced concrete to the inside of the cross-section, causing the internal temperature rise to lag behind that of the outer interface. The maximum temperature of the beam and column cross-sections post-fire decreased non-linearly from the surface to the interior, as clearly indicated by the data.

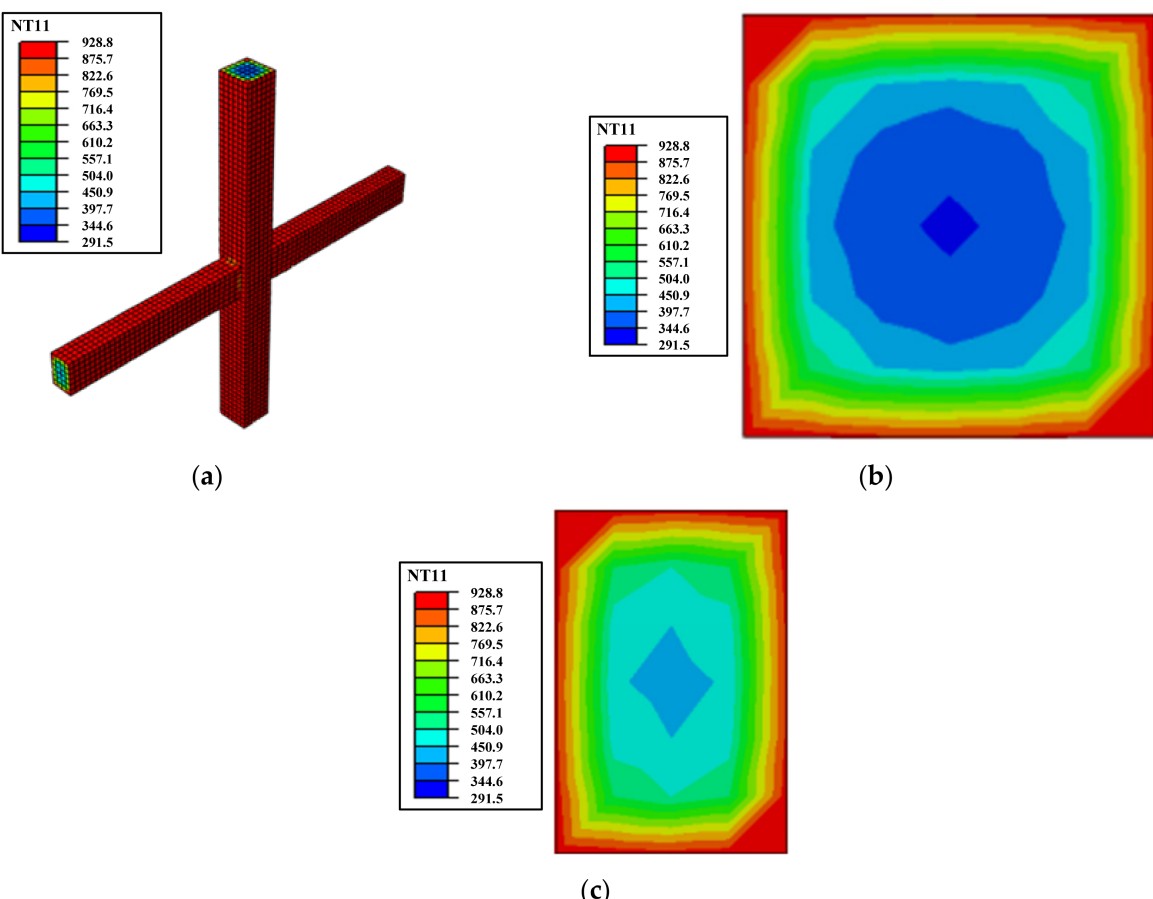

**(a)**                                                                                       **(b)**

**(c)**

**Figure 3.** Maximum temperature field of the joint. (**a**) Axonometric projection of the joint; (**b**) column section; (**c**) beam section.

## 4. Joint Reinforcement after Fire

Carbon fiber composites, composed of over 95% carbon content and exhibiting attributes such as low weight, thinness, exceptional physical and mechanical properties, and robust adhesion, were selected as reinforcement materials for joints post-fire. This study utilized a full bonding method on all four sides for joint reinforcement, with the related methods and dimensions depicted in Figure 4.

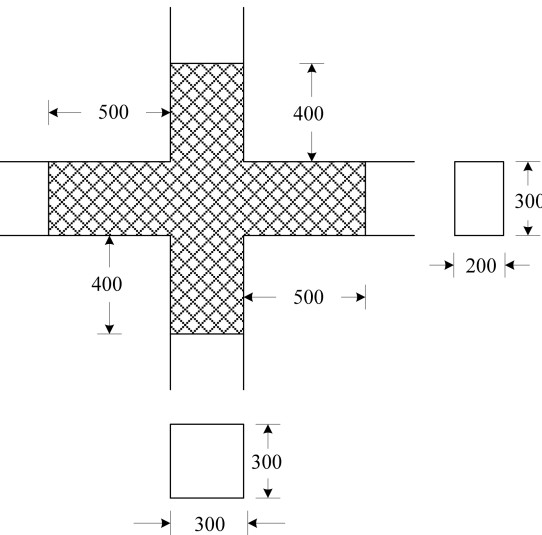

**Figure 4.** Reinforcement and dimensions (unit: mm).

Carbon fiber-reinforced polymer (CFRP) is ideally linear elastic with only ultimate tensile strength and no yield strength; fiber fracture is considered when fiber stress exceeds its tensile strength. This study employed CFRP manufactured by Dezhou Junteng Material Technology Co. Ltd., Dezhou, China, with the primary mechanical property indices presented in Table 3. In finite element calculations, a membrane element is used for CFRP, and a Tie constraint is set for the contact relationship between CFRP and concrete. The bond-slip between them is disregarded, and it is assumed that the bond between the two is strong.

**Table 3.** Mechanical performance of CFRP.

| Model Specification/(g/m²) | Design Thickness/mm | Standard Tensile Strength/MPa | Elastic Modulus/MPa | Elongation/% |
|---|---|---|---|---|
| 300 | 0.167 | 3027 | $2.16 \times 10^5$ | 1.51 |

## 5. Post-Fire Seismic Calculation of Reinforced Joints

### 5.1. Post-Fire Material's Constitutive Relationship

Various studies [12–14,26] showed different constitutive models of steel and concrete in pre- and post-fire conditions. This study adopted the constitutive models with the least error compared to experimental data.

It is widely accepted that the mechanical performance of steel largely recovers after exposure to high temperatures [14,26]. Specifically, the elastic modulus, Poisson's ratio, yield and ultimate stresses, and stress–strain behavior are similar comparable to those at room temperature. Therefore, these properties were assumed to be equivalent to those at room temperature in this study. Figure 5 illustrates the uniaxial hysteretic constitutive model of the reinforcement.

The maximum temperature experienced at various points within the component after exposure to high temperatures varied. The concrete's elastic phase was primarily determined by the material's Young's modulus and Poisson's ratio. As the Poisson's ratio did not change with temperature, it was considered to be the recommended value of 0.2 at room temperature. The room-temperature Young's modulus is calculated through linear interpolation from the table provided in the Standard ISO-834 [30]. The relationship

describing the change in the concrete's secant modulus with temperature after being exposed to high temperature was given by a specific formula:

$$\frac{E_{c,Tm}}{E_c} = \begin{cases} 1.027 - 1.335(\frac{T_m}{1000}) & T_m \leq 200\ ^\circ C \\ 1.335 - 3.371(\frac{T_m}{1000}) + 2.382(\frac{T_m}{1000})^2 & 200\ ^\circ C < T_m \leq 600\ ^\circ C \end{cases} \tag{1}$$

where $E_{c,Tm}$ and $E_c$ represent the elastic modulus of concrete after being exposed to the maximum temperature $T_m$ and that at room temperature.

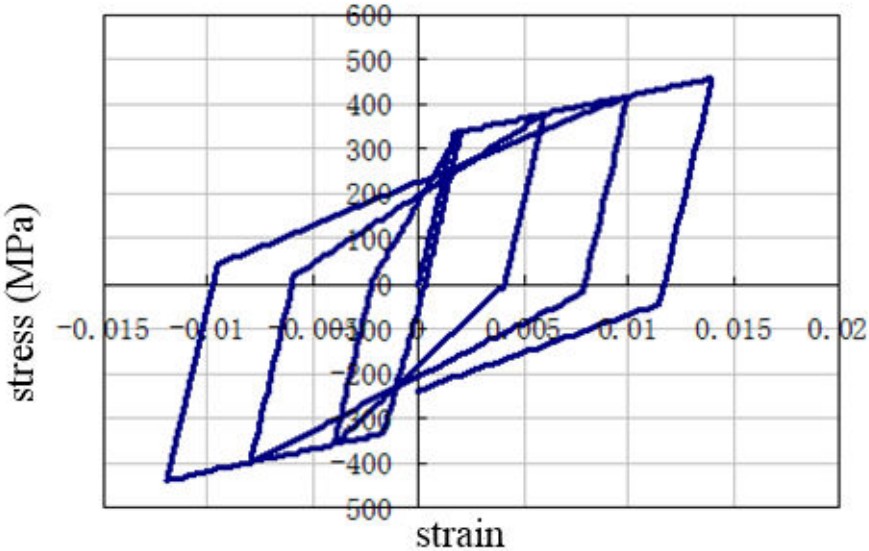

**Figure 5.** Constitutive relationship of the reinforcement.

The concrete's plastic phase was analyzed based on the plastic potential energy equation and yield surface equation. The flow eccentricity, expansion angle, biaxial isobaric yield strength to uniaxial compressive strength ratio, second stress invariant ratio on the tensile and compressive meridian, and viscous parameter were 38, 0.1, 1.16, 2/3, and 0.005, respectively. The concrete's compressive stress–strain relationship after being exposed to high temperatures was determined using a particular model [20–23]:

$$\frac{\sigma}{f_{ck,Tm}} = \begin{cases} 2x - x^2 & 0 < x \leq 1 \\ 1 - \frac{115(x-1)\varepsilon_{0,Tm}}{1 + 5.04 \times 10^{-3}T_m} & 1 < x \leq \varepsilon_{u,Tm}/\varepsilon_{0,Tm} \end{cases} \tag{2}$$

where $x = \varepsilon/\varepsilon_{0,Tm}, \varepsilon_{0,Tm} = \varepsilon_0(1.0 + 2.5 \times 10^{-3}T_m)$, $\varepsilon_{u,Tm} = \varepsilon_u(1.0 + 3.5 \times 10^{-3}T_m)$; $\varepsilon_0$ and $\varepsilon_u$ represent the strain and ultimate strain corresponding to the concrete's peak stress at room temperature; and $\varepsilon_{0,Tm}$ and $\varepsilon_{u,Tm}$ represent the strain and ultimate strain corresponding to the concrete's peak stress after being exposed to the maximum temperature $T_m$.

The concrete's compressive strength experiencing high temperatures was given by another formula [23–25]:

$$\frac{f_{cu,Tm}}{f_{cu}} = \begin{cases} 1 & 0\ ^\circ C \leq T_m \leq 200\ ^\circ C \\ 0.0015(200 - T_m) + 1.0 & 200\ ^\circ C < T_m \leq 500\ ^\circ C \\ 0.003(600 - T_m) + 0.25 & 500\ ^\circ C < T_m \leq 600\ ^\circ C \\ 7.5 \times 10^{-4}(600 - T_m) + 0.25 & 600\ ^\circ C < T_m \leq 800\ ^\circ C \end{cases} \tag{3}$$

where $f_{cu,Tm}$ is the concrete's compressive strength after being exposed to the maximum temperature $T_m$ and $f_{cu}$ is the compressive strength of the concrete at room temperature.

*5.2. Boundary Condition and Loading Pattern*

Referencing the actual boundary conditions of the quasi-static test, a fixed hinge was employed at the column's top end, constraining all degrees of freedom except for U2 and UR1. Likewise, the column's bottom end was fixed to constrain all degrees of freedom except for UR1. A low-cycle reciprocating load was applied to the left and right beams 150 mm away from each end, as depicted in Figure 6. The loading regime is illustrated in Figure 7.

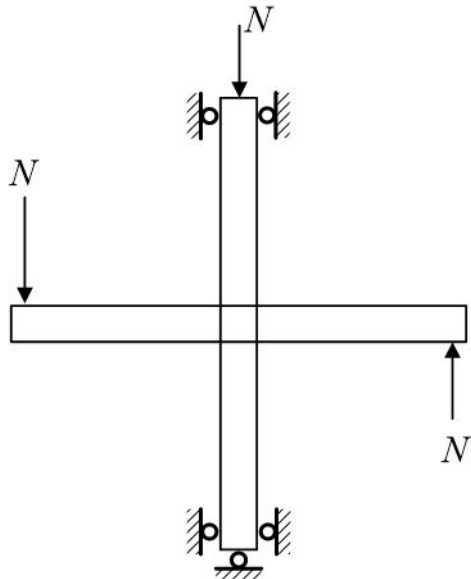

**Figure 6.** Boundary condition.

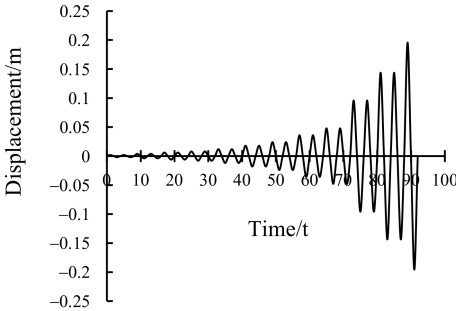

**Figure 7.** Loading regime.

The interface contact in the simulation was treated as follows: an embedded constraint was employed between the concrete and reinforcement, with their bond slip being disregarded. Additionally, the concentrated axial force at the column top and the displacement load point near the beam end were coupled to the upper end face of the column and the surface 150 mm away from the beam end (width: 50 mm) using the point-face coupling constraint.

*5.3. Calculation Results and Analysis*

5.3.1. Hysteretic Curves

The hysteretic curve serves to represent the relationship between a structure's load and deformation under repeated actions, revealing the stiffness degradation, deformation capacity, and energy dissipation capacity of the structure when subjected to reciprocal loading. These curves play a crucial role in determining a structural resilience model and

facilitating non-linear seismic response analysis. Figure 8 provides the hysteretic curves derived from finite element simulations for various CFRP layers.

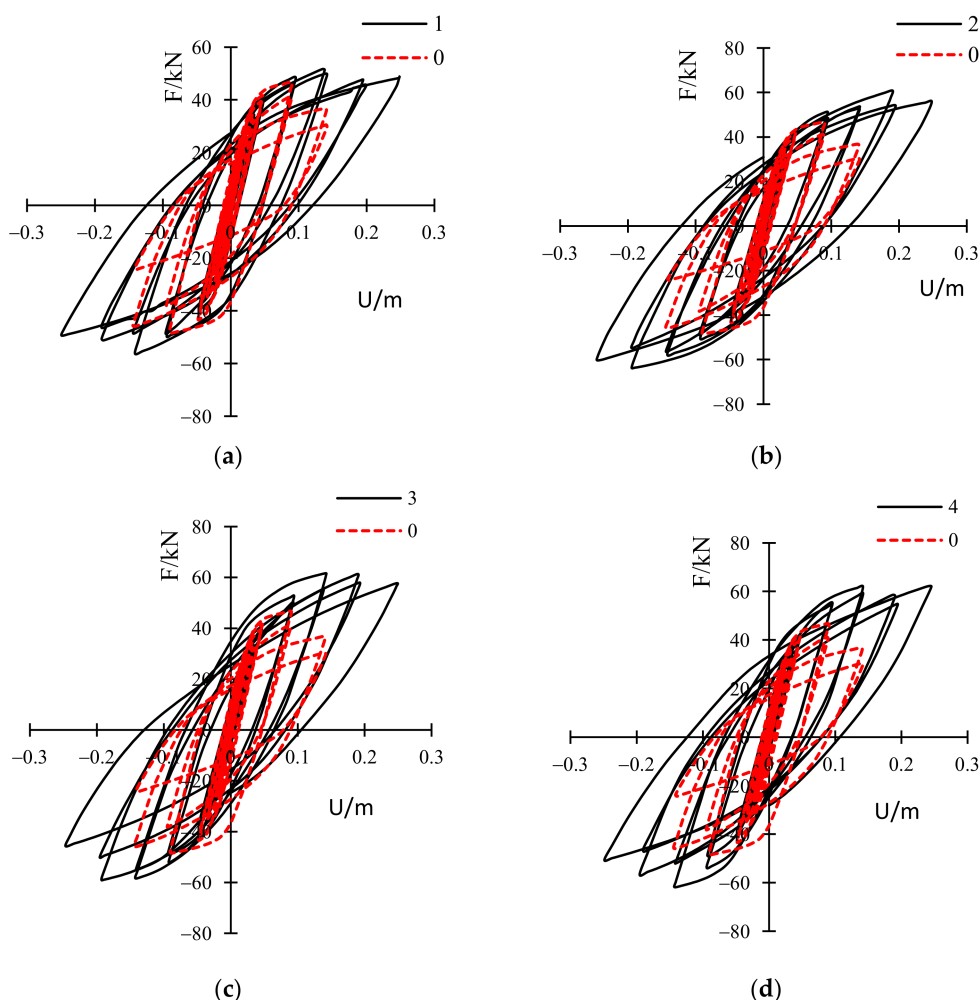

**Figure 8.** Hysteretic curves of the specimen reinforced with different CFRP layers. (**a**) 1 layer; (**b**) 2 layers; (**c**) 3 layers; (**d**) 4 layers.

These curves, notable for their "shuttle" shape after CFRP reinforcement, demonstrated excellent energy dissipation ability. They distinctly divided into three phases. In the elastic phase, the enhanced joint's bearing capacity was marginally and comparably increased relative to the unreinforced specimen. This minimal increase, confirmed by the overlapping hysteretic curves, suggests that the number of CFRP layers exerted a minimal impact on the joint's elastic phase. However, during the plastic phase, the CFRP reinforcement significantly boosted the joint's failure load, ultimate bearing capacity, and failure displacement, thus contributing to improved seismic performance. Intriguingly, as the number of CFRP layers increased, the rate of decrease of the hysteretic curve's descending section also gradually accelerated.

### 5.3.2. Skeleton Curves

The skeleton curve, defining the envelope of the hysteretic curve between load and displacement, informs about the specimen's cracking load, ultimate bearing capacity, and load-deformation relationship. It forms the basis for seismic performance analysis.

Figure 9 depicts the skeleton curves for joints reinforced with varying numbers of CFRP layers. Notably, these curves significantly overlapped during the elastic phase, indicating

that the reinforcement layer number had a minimal influence on the joint's bearing capacity during this phase. Nonetheless, the joint's bearing capacity markedly improved following the elastic phase as the reinforcement layer number increased. However, this enhancement ceased to grow with further increase in the number of reinforcement layers. When 3 and 4 layers were employed, the skeleton curves nearly coincided during the elastic-plastic phase, and the extent of bearing capacity improvement was no longer significant. Compared to unreinforced components, the ductility of reinforced components increased, and the descending sections of the skeleton curve became gentler. Additionally, the descending section's rate of acceleration increased with the number of CFRP layers. This can be attributed to a significant stiffness increase at the beams and columns intersection due to the addition of multiple CFRP layers, which, in turn, could enhance the overall brittleness of the structure.

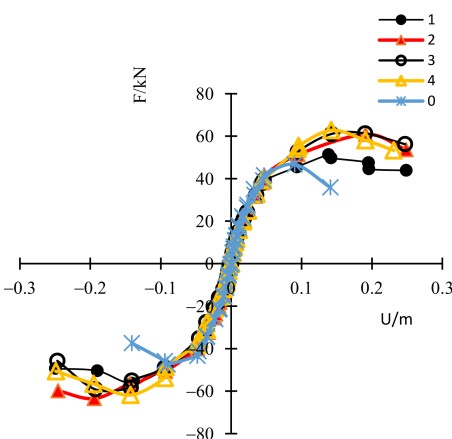

**Figure 9.** Skeleton curves of the specimen reinforced with different CFRP layers.

### 5.3.3. Joint Bearing Capacity of Joints

Table 4 shows the bearing capacity of joints reinforced with different numbers of CFRP layers. As per the table, the yield load, ultimate load, and failure load of the joint increased incrementally with the number of layers. For 1, 2, 3, and 4 layers, the joint's ultimate bearing capacity measured 51,291.2 kN, 60,844 kN, 61,315.1 kN, and 61,816.4 kN, respectively. These values reflect an increase in the joint's load-bearing capacity of 9.8%, 30.3%, 31.3%, and 32.4%, respectively, compared to the unreinforced components. It was clear that the joint's bearing capacity rose with the number of layers, although the growth rate diminished over time. Notably, the reinforcement effect was most pronounced when two reinforcement layers were used, as evidenced by the fastest growth rate.

**Table 4.** Specimen's bearing capacity when reinforced with different layers of CFRP.

| Number of Reinforcement Layers | Yield Load/kN | Ultimate Load/kN | Failure Load/kN | Growth Rate of the Ultimate Load |
|---|---|---|---|---|
| 0 | 35,011.8 | 46,682.4 | 35,011.8 | - |
| 1 | 38,468.4 | 51,291.2 | 43,597.52 | 9.8% |
| 2 | 45,633.375 | 60,844.5 | 51,717.825 | 30.3% |
| 3 | 45,986.325 | 61,315.1 | 52,117.835 | 31.3% |
| 4 | 46,362.3 | 61,816.4 | 52,543.94 | 32.4% |

### 5.3.4. Energy Dissipation

The energy dissipation capacity, which measures a structure or component's ability to absorb energy by undergoing irreversible deformation during repeated loading, is a crucial parameter to evaluate a ductile structure's seismic performance. In this study, we utilized

the equivalent viscous damping coefficient ($h_e$) to analyze the structure's energy dissipation capacity. This coefficient can be calculated based on the hysteresis loop illustrated in Figure 10.

$$h_e = \frac{1}{2\pi} \cdot \frac{S_{ABC} + S_{CDA}}{S_{OBE} + S_{ODF}} \qquad (4)$$

where $S_{ABC}$ is the area enclosed by curves AB, BC, and straight line AC, $S_{CDA}$ is the area enclosed by curves CD, DA, and straight line AC, and $S_{OBE}$ and $S_{ODF}$ are the areas of right triangles OBE and ODF, respectively.

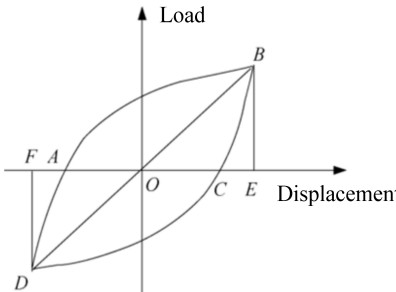

**Figure 10.** Load–displacement hysteretic loop.

Table 5 details the characteristics of joints reinforced with various numbers of CFRP layers. After reinforcement, the joint's energy dissipation capacity showed noticeable improvement. Specifically, for structures reinforced with 1, 2, 3, and 4 layers, the energy dissipation capacity increased by 20.7%, 26.5%, 19.6%, and 11.7%, respectively, when compared to unreinforced structures. However, it is important to note that the increase in energy dissipation capacity was not directly proportional to the number of reinforcement layers. The structure reinforced with two layers showed the highest increase in energy dissipation capacity, indicating a superior reinforcement effect.

**Table 5.** Energy dissipation ability of the specimen strengthened with different CFRP layers.

| Number of Reinforcement Layers | $h_e$ | Growth Rate |
|:---:|:---:|:---:|
| 0 | 0.2106 | - |
| 1 | 0.2542 | 20.7% |
| 2 | 0.2664 | 26.5% |
| 3 | 0.2521 | 19.6% |
| 4 | 0.2351 | 11.7% |

5.3.5. Ductility Analysis

The ductility coefficient can provide a better assessment of a component's seismic performance. In this study, we used the displacement ductility coefficient to measure the ductility of joints.

$$\mu_\Delta = \frac{\Delta_u}{\Delta_y} \qquad (5)$$

where $\Delta_u$ and $\Delta_y$ represent ultimate deformation and yield deformation, respectively.

Table 6 illustrates the ductility of joints reinforced with different numbers of CFRP layers. The yield displacement increased somewhat after reinforcement and remained relatively unchanged thereafter. Meanwhile, the failure displacement first increased and then decreased for structures reinforced with 1, 2, 3, and 4 layers. The ductility coefficient displayed a decreasing trend with an increasing number of reinforcement layers. The joint's ductility reached its peak with a single layer of CFRP, and significantly diminished with four layers of reinforcement. This observation may be attributed to the marked increase

in the stiffness of the beam–column intersection within the joint, and a larger stiffness differential with other parts as the number of CFRP layers increased.

**Table 6.** Specimen's ductility when reinforced with different layers of CFRP.

| Number of Reinforcement Layers | Yield Displacement/mm | Failure Displacement/mm | Ductility Coefficient | Growth Rate of the Ductility |
|---|---|---|---|---|
| 0 | 33 | 137 | 4.15 | - |
| 1 | 53 | 261 | 4.92 | 18.5% |
| 2 | 59 | 268 | 4.55 | 9.4% |
| 3 | 66 | 285 | 4.31 | 3.8% |
| 4 | 67 | 237 | 3.53 | −14.7% |

5.3.6. Stress Distribution

The stress distribution of the structure under various axial compression ratios (0.3, 0.5, 0.7, and 0.8) at peak load is presented in Figure 11. The high-stress zones within the joint expanded as the axial compression ratio increased. At axial compression ratios of 0.3 and 0.5, the peak stress values primarily localized at the beam end where it intersected with the column, and at the joint core, resulting in ductile failure mainly at the beam end. When the ratio increased to 0.5, even though failure still primarily occurred at the beam end, the stress value in the column area significantly increased compared to that at 0.3, suggesting a shift in the joint's failure mode. When the ratios reached 0.7 and 0.8, the stress values in most of the column exceeded those at the end of the beam and the joint core, indicating that joint failure was primarily manifested as the crushing failure of the column. This suggests that the joint's failure mode transitions from plastic damage at the beam end under initial low axial compression ratios to crushing failure of the column under high axial compression ratios.

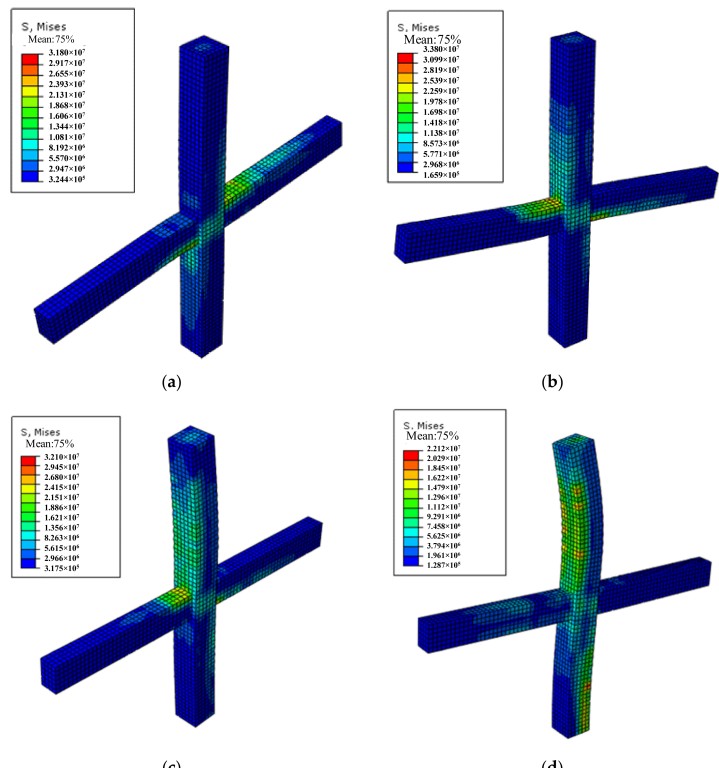

**Figure 11.** Stress distribution of the specimen subjected to different axial compression ratios. (**a**) *n* = 0.3; (**b**) *n* = 0.5; (**c**) *n* = 0.7; (**d**) *n* = 0.8.

## 6. Conclusions

This study presented a numerical simulation of the post-fire seismic performance of reinforced concrete frame joints reinforced with CFRP. This simulation incorporated a fire-exposed temperature field model and a structural mechanics calculation model for the post-fire scenario. References [32–34] presented experimental research on the seismic performance of structures post-fire when strengthened with CFRP. These studies consistently demonstrated improvements in the bearing capacity and ductility of joints, a more robust hysteresis curve, enhanced deformation capacity, and slower degradation of strength and stiffness after reaching the ultimate load, following reinforcement. These consistent experimental phenomena strongly support the reliability of our simulated results. Based on our findings, we concluded the following:

- When the specimen is subjected to fire on all four sides, the temperature field of the beam and column sections displays symmetry with respect to the longitudinal and transverse axes of the corresponding cross-sections. The heat is transferred layer by layer from the outermost reinforced concrete layer inward, resulting in a delayed rise of the internal temperature compared to the outer interface. After exposure to fire, the maximum temperature of the beam and column cross-sections decreases non-linearly from the surface towards the interior.
- In the elastic phase post-fire, the bearing capacity of the reinforced joint remains largely comparable to that of the unreinforced joints. However, the ductility, energy dissipation capacity, and ultimate bearing capacity of the reinforced joint demonstrate significant improvements, though this enhancement does not continue with an increasing number of reinforcement layers. The use of two reinforcement layers results in an energy dissipation capacity and ultimate bearing capacity increase of 26.5% and 30.3%, respectively, showcasing an excellent reinforcement effect. These results suggest that the repaired joints effectively regain their original strength and stiffness.
- Analysis of the joint's stress distribution reveals that as the axial compression ratio increases, the high-stress zones at the joint expand. The failure mode also transitions from plastic damage at the joint's beam end under low axial compression ratios to the column's crushing failure under high axial compression ratios.

Our findings indicate that the use of CFRP is a viable method for strengthening reinforced concrete frame joints post-fire. This approach enhances the mechanical properties of the structure and aligns with sustainable practices of continuous use and development. Future research could explore the impact of the bonding method and the dimensions of CFRP on the mechanical properties of the structure. This could further promote the application of the CFRP reinforcement method in the restoration of building structures.

**Author Contributions:** J.W. and Y.S. conceived the simulation; X.Y. performed the simulation and analyzed the data; F.L. and Z.L. wrote the paper. All authors have read and agreed to the published version of the manuscript.

**Funding:** The paper was funded by the Key Scientific Research Project Plan of Colleges and Universities in Henan Province, China (Grant No. 20ZX009).

**Institutional Review Board Statement:** Not applicable.

**Informed Consent Statement:** Not applicable.

**Data Availability Statement:** Not applicable.

**Conflicts of Interest:** The authors declare no conflict of interest.

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
