# Peer review of "Post-Fire Seismic Property of Reinforced Concrete Frame Joints with Carbon Fiber-Reinforced Polymer Using Numerical Analysis"

_fire, doi:10.3390/fire6050205_

Round 1

Reviewer 1 Report

I have reviewed the paper “Post-fire seismic property of reinforced concrete frame joints with carbon fiber reinforced polymer using numerical analysis” by Jinyan Wang, Xingchao Yue, Fuli Li, Yuzhou Sun and Ziqi Li. The topic fits well within those of the journal and the work is technical sound. It may be of interest for both researchers and professional engineers. Therefore, this reviewer is in the opinion that the paper can be accepted after changes are made by the authors according to the following suggestions.

1. Introduction

Literature review needs to be improved. Some recent work referring to the seismic retrofitting of fire-damaged RC framed buildings by means of innovative techniques (e.g. base-isolation) should be added. Useful information can be found in the following papers:

Mazza, F. Residual seismic load capacity of fire-damaged rubber bearings of r.c. base-isolated buildings. Engineering Failure Analysis 2017, 79, 951-970.

Lucon, M., Baragatti, P.Possidente, L., Tondini, N. Experimental fire response of seismic elastomeric bearings. Engineering Structures 2022, 254, 113806.

2. Specimen Design

Table 1. What does it mean C16? What does it mean A8?

Table 2. Please clarify the first column.

3. Simulation of Temperature Field

Figures 3b and 3c. Please clarify the asymmetric distribution of temperature (e.g. the four corners do not have the same temperature).

Figures 3b and 3c. It is not clear if the temperature distribution refers to the time corresponding to the maximum temperature of the time-temperature curve (i.e. t= 60 min) or to the end of the cooling phase. Please clarify it.

4. Joint Reinforcement after Fire

Table 3. Please clarify the parameter reported in the first column.

5.3. Calculation Results and Analysis

Please specify the normalized axial load corresponding to the hysteretic curves (Figure 8), skeleton curves (Figure 9), bearing capacity (Table 4), energy dissipation (Table 5) and ductility (Table 6).

Figure 10. Please specify the time of the ISO834 time-temperature curve.

Minor spell check required

Reviewer 2 Report

This manuscript presents numerical simulations of post-fire seismic behaviour of reinforced concrete beam-column joints with CFRP. Numerical models were established which considered the effect of high temperature on material properties of steel reinforcement and concrete. The study is of interest, but the manuscript is not suitable for publication in the present form. Comments can be found below.

1. The English language should be carefully polished.

2. Numerical results in the present study are not validated against test data, which makes the reliability of numerical models in question.

3. In addition to the global behaviour and failure model, detailed analyses should also be provided for the plastic hinge and joint core.

4. The manuscript is not properly formatted, such as the symbols in equations and the conclusion.

The English language should be improved. 

Reviewer 3 Report

2) The figures are illegible; please rectify this problem.

3) The writers must highlight the value of their work in the opening section. Then, they should explain why this study is necessary and contrast the drawbacks of earlier studies.

4) It must be plain to us why the joint was analyzed 60 minutes after the fire was exposed, as opposed to 15 minutes.

5) Describe how variations in the temperature field can be used to determine the mechanical properties of concrete materials.

1) According to this reviewer, the manuscript's wording may use some refinement.

Reviewer 4 Report

The study is focused on the post-fire seismic response of RC joints retrofitted via carbon fiber reinforced polymers. The investigation is carried out via numerical models implementing the degraded structural behaviour after fire and the effect of four retrofit strategies. The following comments should be addressed.

Introduction: An outline is required to explain the reader the contents of the different sections and better clarify the methodology adopted in the study. In addition, the state-of-the-art should include previous studies on the response of RC structures under fire and their post-fire seismic response. This state-of-the-art should consider studies related to the international context. It seems only studies related to the Chinese context are included in this version of the manuscript.

The result discussion should include a comparison between pre-fire and post-fire response of the joint without retrofitting. The response of the un-retrofitted joint in pre-fire condition can allow the reader understanding the effectiveness of the retrofit strategy to restore the initial structural behaviour. In addition, the author should present comparisons between the different constitutive models of steel and concrete related to pre and post fire conditions.

The authors should include a judgemental explanation on why more than 3 layers induce a negligible influence on backbone curves of the retrofitted joint.

L232 The authors should describe how the equivalent viscous damping coefficient is computed (e.g. 10.1002/eqe.2257 or 10.1016/j.soildyn.2021.106829).

L63 Correct the sentence “seismic experiments of …”

L47 Please correct the sentence “and some progress has achievements have been madeattained in strengthening room-temperature reinforced concrete structures at room temperature”.

L130 - references are required.

Minor improvements (in clarity of the sentences) are required.

Round 2

Reviewer 1 Report

The changes and additions introduced in the revised version of the manuscript met the requests of the Reviewer. 

In view of this, in my opinion the revised manuscript can be accepted as is for publication.

Author Response

Thank you for your comments.

Reviewer 3 Report

Accept in present form

 English language fine.

Author Response

Thank you for your comments.